# In-Situ Fabrication of g-C_3_N_4_/ZnO Nanocomposites for Photocatalytic Degradation of Methylene Blue: Synthesis Procedure Does Matter

**DOI:** 10.3390/nano9020215

**Published:** 2019-02-06

**Authors:** Shengqiang Zhang, Changsheng Su, Hang Ren, Mengli Li, Longfeng Zhu, Shuang Ge, Min Wang, Zulei Zhang, Lei Li, Xuebo Cao

**Affiliations:** 1College of Biological, Chemical Sciences and Engineering, Jiaxing University, 118 Jiahang Road, Jiaxing 314001, China; zsq@yunyaochina.cn (S.Z.); rh136@outlook.com (H.R.); zhulf1988@mail.zjxu.edu.cn (L.Z.); geshuang8@126.com (S.G.); jerry3641172@126.com (Z.Z.); 2Department of Chemical and Biomolecular Engineering, University of Notre Dame, IN 46556, USA; changsheng.su@cummins.com; 3State Key Laboratory of High Performance Ceramics and Superfine Microstructure, Shanghai Institute of Ceramics, Chinese Academy of Sciences, 1295 Ding-Xi Road, Shanghai 200050, China; xiaomindafeng@163.com

**Keywords:** ZnO nanorods, dye photodegradation, graphitic carbon nitride, nanocomposites, in situ synthesis

## Abstract

The nanocomposite preparation procedure plays an important role in achieving a well-established heterostructured junction, and hence, an optimized photocatalytic activity. In this study, a series of g-C_3_N_4_/ZnO nanocomposites were prepared through two distinct procedures of a low-cost, environmentally-friendly, in-situ fabrication process, with urea and zinc acetate being the only precursor materials. The physicochemical properties of synthesized g-C_3_N_4_/ZnO composites were mainly characterized by XRD, UV–VIS diffuse reflectance spectroscopy (DRS), N_2_ adsorption-desorption, FTIR, TEM, and SEM. These nanocomposites’ photocatalytic properties were evaluated in methylene blue (MB) dye photodecomposition under UV and sunlight irradiation. Interestingly, compared with ZnO nanorods, g-C_3_N_4_/ZnO nanocomposites (*x*:1, obtained from urea and ZnO nanorods) exhibited weak photocatalytic activity likely due to a “shading effect”, while nanocomposites (*x*:1 CN, made from g-C_3_N_4_ and zinc acetate) showed enhanced photocatalytic activity that can be ascribed to the effective establishment of heterojunctions. A kinetics study showed that a maximum reaction rate constant of 0.1862 min^-1^ can be achieved under solar light illumination, which is two times higher than that of bare ZnO nanorods. The photocatalytic mechanism was revealed by determining reactive species through adding a series of scavengers. It suggested that reactive ●O_2_^−^ and h^+^ radicals played a major role in promoting dye photodegradation.

## 1. Introduction

Solar energy is believed to be the most abundant source of sustainable and clean energy [1]. Photocatalysis is a developing green technology that can utilize solar energy for a wide range of applications, such as the decomposition of various organic compounds in water purification processes [2,3,4,5], hydrogen production and CO_2_ photoreduction in energy conversion [6,7,8], and bacterial inactivation in medicine [9]. Among well-known semiconductor photocatalysts, ZnO is promising because of its remarkable properties, such as physicochemical/thermal stability, low-cost, high redox potential, and electron mobility [10,11,12]. However, its incomplete visible light absorption, poor charge separation, and susceptibility to photo-corrosion have seriously hindered its practical applications. Extensive attention has therefore been made to enhance the photocatalytic activity and stability of ZnO based photocatalysts via heteroatom doping [13,14,15,16], novel metal deposition [17,18], or coupling with narrow band gap semiconductors [19,20,21].

Recently, a light non-metal semiconductor graphitic carbon nitride (g-C_3_N_4_) has attracted much interest due to its unique electronic structure, remarkable chemical stability, visible light activity, and cost-effective features [22,23]. As a typical analogue of graphite and new photocatalyst, g-C_3_N_4_ possesses a bandgap of ~2.7 eV, which enables it to be excited by visible light [24]. However, like many other single-component photocatalysts, its quantum efficiency is negatively affected by the fast recombination of photo-generated holes and electrons [25]. Therefore, it still remains a challenge to obtain highly stable and active g-C_3_N_4_ based photocatalysts.

Coupling g-C_3_N_4_ with ZnO to form heterostructured nanocomposites presents a novel and feasible route to enhance light sensitivity and charge carrier separation [26]. The improved adsorption of visible light and photocatalytic activity of g-C_3_N_4_/ZnO nanocomposites could be attributed to the establishment of a heterojunction between g-C_3_N_4_ and ZnO [26]. In their pioneering studies, Wang et al. [27] and Liu et al. [28] independently synthesized g-C_3_N_4_ and ZnO heterostructured composites through a two-step chemisorption method [27] and a ball milling method [28]. In a later report, Zhu et al. [29] were the first to introduce the preparation of g-C_3_N_4_/C-doped ZnO composites through a calcination method. More recently, Wang et al. constructed core-shell g-C_3_N_4_/ZnO nanocomposites as a photoanode via a reflux method [10], demonstrating enhanced visible-light-irradiated photoelectrocatalytic (PEC) performance for phenol decomposition. One dimensional ZnO nanorods were successfully coated with g-C_3_N_4_ through reflux and vapor condensation procedures [30], and the photocurrent density of those fabricated 1-D ZnO/g-C_3_N_4_ composites was approximately 0.12 mA∙cm^−2^.

The nanocomposite preparation procedure plays a crucial role in achieving the well-established heterostructured junction between g-C_3_N_4_ and ZnO, and hence, an optimized photocatalytic activity [29]. Nevertheless, the synthesis of g-C_3_N_4_/ZnO nanocomposites usually requires the use of environmentally-unfriendly organic solvents or involves an undesired multi-step synthesis technique. In previous reported studies, the synthesis of ZnO nanorods and g-C_3_N_4_ were introduced through a facile calcination process [30]. This low-cost, environmentally-friendly method leads to the formation of g-C_3_N_4_/ZnO nanocomposites through a simple and effective in-situ process. Remarkably, we used urea and zinc acetate as the only precursor materials to synthesize g-C_3_N_4_/ZnO nanocomposites with a g-C_3_N_4_:ZnO weight ratio of *x*:1 (*x* = 0.05, 0.1, 0.2 and 0.3) through a two-step in-situ calcination method. Particularly, as illustrated below, g-C_3_N_4_/ZnO nanocomposites were prepared by using zinc acetate and urea, respectively, as the starting material of the in-situ synthesis process. The specific surface area of these samples increased with the increase of the g-C_3_N_4_ weight ratio, and the results are shown in Appendix A and Appendix A.

We explore here the effect the synthesis procedure has on the physiochemistry, photoelectrochemistry (PEC) properties, and photodegradation activities of g-C_3_N_4_/ZnO nanocomposites (*x*:1 vs. *x*:1 CN) prepared by an in-situ calcination route. Interestingly, in comparison with ZnO nanorods, nanocomposites (*x*:1) exhibited weak photocatalytic activity probably due to the “shading effect” [31,32], while nanocomposites (*x*:1 CN) showed enhanced photodegradation activity due to the effective establishment of heterojunctions [33]. This work is believed to be the first attempt in the field to investigate the effect of g-C_3_N_4_/ZnO synthesis procedure by in-situ calcination methods. The in-situ calcination approach benefits from low-cost and potential to scale for industrial applications.

## 2. Materials and Methods

### 2.1. Catalyst Preparation

Urea and zinc acetate were supplied by Sinopharm Chemical Reagent Co., Ltd (Shanghai, China). Methylene blue, ammonium oxalate (AO), and isopropanol (IPA) were purchased from Aladdin reagent Co., Ltd (Shanghai, China). All the chemicals were analytical grade and used as received without any further purification.

ZnO nanorods were synthesized by a one-step method described elsewhere [34]. Briefly, 2.94 g zinc acetate dehydrate was added in an agate mortar and grinded for about 30 min. Finally, this grinded powder was put in an alumina crucible and annealed at 350 °C for 3 h with a ramp rate of 5 °C/min. To obtain the g-C_3_N_4_ powder, an appropriate dose of urea in an alumina crucible was thermally decomposed at 550 °C for 2 h in a muffle furnace (Nabertherm L15, Lilienthal, German) under static air, as described in our previous studies [35].

Scheme 1 illustrates the details of two distinct g-C_3_N_4_/ZnO synthesis paths. As for *x*:1 samples (*x*:1 represents weight ratio of g-C_3_N_4_ and ZnO), synthesized ZnO nanorods were grinded and mixed with various amounts of urea, and then annealed with a cover at 550 °C for 2 h.

The *x*:1 CN samples were obtained by grinding the necessary amounts of g-C_3_N_4_ powder mixed with zinc acetate in a quartz mortar. The mixture was then calcinated in a muffle furnace at 350 °C for 3 h and then cooled to room temperature naturally.

### 2.2. Catalytic Evaluation

Methylene blue (MB, 1 × 10^−5^ mol/L) photocatalytic degradation was typically conducted in a quartz tube under ultraviolet light (Philips, 18 W) or sunlight (place location: 120.76° in longitude, and 30.77° in latitude), and 50 mg of photocatalyst was dispersed in 100 mL of MB solution each time. The resulting solution was stirred in the dark for 30 min to ensure that the mixture had reached an adsorption and desorption equilibrium, and then the solution was illuminated under stirring. Five milliliters of solution was sampled from the tube at regular intervals and analyzed by UV–visible spectrophotometry (Shimadzu UV-2450). During the stability test, the suspension was centrifuged and washed with deionized water several times for each run. Finally, the powder photocatalyst was collected and dried in an oven in air at 60 °C overnight prior to being used for the next run.

### 2.3. Characterization

These as-synthesized nanocomposites were studied by various techniques. X-ray powder diffraction (XRD) patterns were collected by a German Bruker D8 Focus Powder (Fremont, CA, USA) using Cu Kα radiation. The scanning speed was setted to 4°/min. The specific surface area of these samples was calculated by the Brunauer–Emmett–Teller (BET) method and characterized by sorption using a Micromeritics Tristar 2000 instrument (Atlanta, GA, USA) at 77 K. Transmission electron microscopy (TEM, JEM-2100F, Tokyo, Japan) and selected area electron diffraction (SAED) patterns were operated at 200 kV. Field emission scanning electron microscopy (FE-SEM) images were obtained with a Hitachi S-4800 (Tokyo, Japan). UV–visible diffuse reflectance spectroscopy (DRS) was acquired on a UV-3101 PC Shimadzu spectroscope (Kyoto, Japan) at room temperature, and BaSO_4_ was chosen as reference material. The photoluminescence (PL) spectra were recorded on a Hitachi F-4500 fluorescence spectrometer (Tokyo, Japan). X-ray photoelectron spectroscopy (XPS) was taken on a Thermo Scientific ESCALAB 250 spectrometer (Waltham, MA, USA) with a monochromatic X-ray line source of Al Kα radiation. All the binding energies were calibrated internally by setting C 1s to 284.6 eV. The surface molecular structural was studied by Fourier transform infrared (FTIR) spectroscopy (470 FI-IR, Medison, WI, USA).

## 3. Results and Discussion

### 3.1. Physicochemical Properties

The XRD analysis in Figure 1 shows patterns of g-C_3_N_4_, ZnO, and the nanocomposites synthesized by two distinct in-situ synthesis procedures with different g-C_3_N_4_:ZnO weight ratios. The nanocomposites prepared with g-C_3_N_4_ as the starting precursor are denoted as *x*:1 CN. The distinct peak at 2 theta of 13.1° for g-C_3_N_4_ belonged to the (100), corresponding to in-plane packing motif, and the other one at 27.4° originated from the (002) stacking structure [36]. Likewise, the distinct diffraction peaks at 2 theta of 31.8°, 34.0°, 36.3°, 47.5°, 56.9°, 62.9°^o^ and 68.0° for pure ZnO indicated its hexagonal structure (JCPDS No. 36-1451). When using zinc acetate as the starting precursor, the g-C_3_N_4_ phase could not be detected until the g-C_3_N_4_ weight ratio in nanocomposite was greater than 0.1:1. In contrast, there was no evidence of g-C_3_N_4_ crystalline formation, even with the increase of g-C_3_N_4_ content when using urea as the starting precursor, which can be ascribed to the high dispersion of g-C_3_N_4_.

SEM images depicted in Figure 2 are ZnO nanorods, g-C_3_N_4_/ZnO (0.1:1), and g-C_3_N_4_/ZnO (0.1:1 CN), respectively. The morphology of as-prepared ZnO nanorods is shown in Figure 2a. The length of these nanorods was in the magnitude of 1 µm, with the diameter of 30–100 nm. After mixing with urea and followed with in-situ calcination, the morphology of formed nanocomposites g-C_3_N_4_/ZnO (*x*:1) showed that ZnO nanorods were wrapped with g-C_3_N_4_ resulting from urea thermopolymerization (Figure 2b). In contrast, the morphology of g-C_3_N_4_/ZnO (0.1:1 CN) showed only moderate coverage of g-C_3_N_4_ over ZnO nanorods.

The morphology of ZnO crystalline and g-C_3_N_4_/ZnO composites was also investigated by TEM. As can be seen in Figure 3a, the as-prepared ZnO nanorods showed a characteristic rod-shaped crystalline structure, and the corresponding hexagonal crystalline plane (0001) [34] was demonstrated by SAED shown in the inset of Figure 3a. TEM and HRTEM images of g-C_3_N_4_/ZnO (*x*:1) composites are present in Figure 3c and 3d, suggesting that ZnO nanorods were coarsened after the second step calcination with the presence of urea, and the nanorods were completely covered with g-C_3_N_4_ even at a g-C_3_N_4_:ZnO ratio of 0.1:1, which is in accordance with previous SEM observations. In comparison, Figure 3b shows the TEM image of g-C_3_N_4_/ZnO (*x*:1 CN) nanocomposites, where ZnO nanorods were slightly covered with g-C_3_N_4_. The EDX images shown in Figure 3e also suggests the homogeneous interaction of g-C_3_N_4_ with ZnO in the case of g-C_3_N_4_/ZnO (*x*:1 CN) composites. After 2 h of ultrasonic treatment, there was no evident change in the morphology of g-C_3_N_4_/ZnO (*x*:1 CN) composites, suggesting a strong combination of g-C_3_N_4_/ZnO which is shown in Appendix A. The strong interaction between g-C_3_N_4_ and ZnO is believed to be critical for the efficient formation of heterojunction [27]. The functional groups of ZnO, g-C_3_N_4_ and their nanocomposites were characterized by FTIR as presented in Figure 4. The peaks at 1637 cm^−1^ and 1243 cm^−1^ are attributed to C=N and C–N stretching in g-C_3_N_4_ [3], while the peak at 807 cm^−1^ corresponds to the plane breathing vibration [19]. The peaks at 1406, 1450, and 1560 cm^-1^ might be due to heptazine-derived repeating units [10,37].

DRS was performed to study the optical adsorption properties in the range of 200–800 nm for as-prepared ZnO nanorods, g-C_3_N_4_, and two distinct series of g-C_3_N_4_/ZnO composite (*x*:1, and *x*:1 CN) samples. As can be seen in Figure 5a and 5b, g-C_3_N_4_/ZnO nanocomposites prepared by both routes showed good light absorption, signifying their UV and sunlight-induced catalytic activity. Evidently, the absorption edges of all nanocomposites had red-shifted and demonstrated significantly enhanced absorption between 380 nm and 800 nm, when compared with that of bare ZnO nanorods. The g-C_3_N_4_/ZnO nanocomposites prepared by both routes showed that the absorbance gradually increased with the increase of g-C_3_N_4_ weight ratio, mainly resulting from a relatively narrow bandgap of g-C_3_N_4_ which is demonstrated in Figure 5c. The bandgap energy of ZnO and g-C_3_N_4_ can be calculated using the Kubelka–Munk formula, (αhv)^n^=k(hν-E_g_), where E_g_ is the band gap energy [38]. The extrapolated intercept in Figure 5c gives the corresponding E_g_ value of 2.88 and 3.07 eV for g-C_3_N_4_ and ZnO nanorods, respectively.

Photoluminescence (PL) spectroscopy was performed to determine the efficiency of interfacial charge carrier separation in the nanocomposite photocatalysts. It is known that photoluminescence emissions on semiconductors originate from the radiative recombination of photoelectrons and holes [34]. As shown in Figure 5d, bare g-C_3_N_4_ had a broad PL peak centered at approximately 480 nm. However, the PL results of ZnO and g-C_3_N_4_/ZnO showed emissions of broad violet (~380 nm) and narrow green-yellow bands (~560 nm). ZnO normally exhibits luminescence in the visible spectral range because of intrinsic or extrinsic defects [39,40]. Previous studies [40] suggest that the defects formed during calcination and can be ascribed to an oxygen vacancy and interstitial oxygen. This was confirmed by XPS as described below. The impurity levels could enhance the photogenerated charge carrier separation in photocatalysts. Therefore, the oxygen defects in ZnO based semiconductors could act as active sites, since the redox reactions usually occur on the surface of the photocatalyst. The PL emission intensities decreased with the addition of g-C_3_N_4_ for both cases of g-C_3_N_4_/ZnO (0.1:1) and g-C_3_N_4_/ZnO (0.1:1 CN), which is an indication that the recombination of charge carriers in ZnO had been effectively inhibited after the formation of heterojunction structures between g-C_3_N_4_ and ZnO. The g-C_3_N_4_/ZnO (0.1:1 CN) composite had the lowest emission intensity, implying its high photocatalytic activity. The presence of g-C_3_N_4_ lead to promoted electron transfer in the nanocomposites.

XPS was conducted to better understand surface species of ZnO nanorods and the nanocomposites. Figure 6a demonstrates that the C 1s spectrum of the composite can be fitted with two peaks at 284.6 eV and 288.4 eV, which correspond to the surface adventitious carbon and the sp^2^-bonded C in N=C–N coordination [41]. Unsurprisingly, only adventitious carbon with a peak at 284.6 eV existed in ZnO nanorods. The binding energy values at 532.5 eV, 531.5 eV, and 530.1 eV in O 1s spectra were observed in the heterostructured samples, while three peaks at around 532.4 eV, 531.4 eV and 530 eV existed in ZnO. These peaks can be ascribed to chemisorbed (O_C_) oxygen species, oxygen in deficient regions (O_V_), and O_2_^2−^ species in lattice (O_L_) [42]. It is interesting to find that chemisorbed oxygen species also presented in g-C_3_N_4_/ZnO nanocomposites. These oxygen species are believed to be associated with surface defects [34], implying that g-C_3_N_4_/ZnO (0.1:1 CN) composite has a high amount of surface defects. Many researchers [25,43] suggest that the differences of photocatalytic activity for different semiconductors result from varying oxygen defect concentrations and types. Furthermore, the oxygen species signal at 532.5 eV was caused by a surface hydroxyl functional group. The N 1s spectrum in Figure 6c can be classified into two peaks. The peak centered at 398.6 eV was attributed to the sp^2^-hybridized nitrogen atoms in C=N–C groups [44], while the other one at 400.0 eV was associated with the tertiary nitrogen H–N–C_2_ or N–C_3_ groups [45]. The Zn 2p spectra were deconvoluted into two peaks of 1021.0 and 1044.0 eV as shown in Figure 6d, corresponding to Zn 2p_3/2_ and 2p_1/2_, respectively. Compared with pure ZnO, the binding energy of O 1s of g-C_3_N_4_/ZnO (0.1:1 CN) exhibited a positive shift, while Zn 2p showed a negative shift, suggesting possible chemical bonding formation between ZnO and g-C_3_N_4_ in the nanocomposite. This interaction was beneficial to promote the charge carrier transfer, and thus enhance photocatalytic activity, which could be further demonstrated by following photocatalytic performance results.

### 3.2. Photocatalytic Performance

The photocatalytic degradation results are shown in Figure 7. No detected degradation was observed without catalyst under light irradiation, which means that the MB was quite stable under this condition. It was found that all nanocomposites (both *x*:1 and *x*:1 CN) exhibited weakened photocatalytic activity under UV light irradiation (Figure 7a,c) compared to that of ZnO. This might have resulted from an enhanced PL phenomenon in g-C_3_N_4_ and will be further investigated in future studies. For g-C_3_N_4_/ZnO (*x*:1) nanocomposites, the photodegradation activity continued to improve with the increase of g-C_3_N_4_ weight ratio under sunlight irradiation, although they exhibited less activity than that of bare ZnO nanorods. This could be due to the “shading effect” [32], since ZnO nanorods were wrapped by g-C_3_N_4_ as can be seen from SEM and TEM images. This would result in only the outer g-C_3_N_4_ being activated by photons and catalytically active during MB photodegradation. In contrast, g-C_3_N_4_/ZnO (0.1:1 CN) showed the best photocatalytic activity among all g-C_3_N_4_/ZnO (*x*:1 CN) nanocomposites, which might have resulted from its heterojunction structure leading to low recombination of charge barriers and sufficient active sites. Furthermore, compared to pristine ZnO nanorods, all nanocomposites (both *x*:1 and *x*:1 CN) showed enhanced activity under sunlight irradiation. As shown in Figure 7e, the photocatalytic reaction under sunlight was fit to pseudo-first order kinetics, where the reaction rate constant (k) could be calculated by rate law, −ln(C/C_0_) = kt, with C and C_0_ being the concentrations of MB. The rate constants were calculated as 0.0901 and 0.1862 min^−1^ for ZnO and g-C_3_N_4_/ZnO (0.1:1 CN), respectively. The stability of g-C_3_N_4_/ZnO (0.1:1 CN) was also evaluated (Figure 7f), and there was negligible degradation on the catalytic activity after four repeated cycles, indicating a good promise to potential industrial applications.

### 3.3. Photocatalytic Mechanism

To find out the major active species for the photocatalytic oxidation, several scavengers were added in the photocatalytic system individually to trap and remove corresponding active species. It was reported that ammonium oxalate (AO), isopropanol (IPA), and N_2_ could act as effective scavengers to holes (h^+^), hydroxyl radical (●OH), and superoxide radical (●O_2_^−^) [46,47,48,49,50,51,52,53,54,55,56,57,58,59]. As shown in Figure 8, the photodegradation activity of the g-C_3_N_4_/ZnO (0.1:1 CN) sample had a dramatic decrease with the addition of AO and N_2_ under light irradiation, suggesting that both holes and ●O_2_^−^ are the main oxidative species in a MB degradation reaction. Moreover, the photocatalytic activity decrease was greater in the presence of N_2_, indicating that the ●O_2_^−^ species was more active than holes during the photodegradation process. On the other hand, the introduction of IPA did not have a noticeable impact on the photocatalytic activity, implying that ●OH did not play a significant role in the reaction as an oxidative species.

## 4. Conclusions

A series of g-C_3_N_4_/ZnO nanocomposites were prepared through two distinct procedures of the in-situ fabrication process. In comparison with ZnO nanorods, g-C_3_N_4_/ZnO nanocomposites (*x*:1) exhibited weak photocatalytic activity likely due to a “shading effect”, while nanocomposites (*x*:1 CN) showed enhanced photocatalytic activity that can be ascribed to the effective establishment of heterojunctions. Moreover, g-C_3_N_4_/ZnO nanocomposites (*x*:1 CN) demonstrated high stability, retaining its initial activity after repeated cycles. This work reveals a facile two-step in-situ synthesis method to prepare g-C_3_N_4_ based hybrid materials that are promising for industrial scale applications.

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
