# Peer review of "In-Situ Fabrication of g-C3N4/ZnO Nanocomposites for Photocatalytic Degradation of Methylene Blue: Synthesis Procedure Does Matter"

_nanomaterials, 2019, doi:10.3390/nano9020215_

Round 1
Reviewer 1 Report
This manuscript is about the fabrication of g-C3N4/ZnO composites and the estimation of the photocatalitic activities. The results represent that the 0.1:1 CN prepared by urea starting process shows good performance in MB decompositions. The results are interesting but there are some questions as follow. I think it will be revised as comments.
About Fig.4, the peaks of the composites are not clear because the ratio of g-C3N4 is not much. I cannot find the shift from Fig. 4. They describe the relationship between OH group and photoreactivity, but the estimation of FTIR spectra is not enough. Please explain carefully.
In Fig.3, they wrote that ZnO nanorods were wrapped. However, it is not clear. I think the ZnO nanorods became small and changed to particle shape. Please estimate supecific surface area of each samples. It is important to compare the photocatalitic activites.
About PL, which is the exitation wavelength? I thinck the shapes of the PL spectra of ZnO and composites are strange. Please check the excitation wavelenght.
In XPS spectra, the baselines of O1s and Zn2p look like incollect. In O1s, there are three peaks as same as those of the coposites.
The desctription of the Fig.7 is not enough. Please add the information.
The reason why 0.1:1 CN samples are best photoreactivity is not clear. Please discuss more based on the results.
Author Response
This manuscript is about the fabrication of g-C3N4/ZnO composites and the estimation of the photocatalitic activities. The results represent that the 0.1:1 CN prepared by urea starting process shows good performance in MB decompositions. The results are interesting but there are some questions as follow. I think it will be revised as comments.
Response: Thank you very much for your kind comment.
About Fig.4, the peaks of the composites are not clear because the ratio of g-C3N4 is not much. I cannot find the shift from Fig. 4. They describe the relationship between OH group and photoreactivity, but the estimation of FTIR spectra is not enough. Please explain carefully.
Response: Thank you very much for your comment and question. The shift at around 1390 cm-1 could be found in magnifying picture. To increase the accuracy of description, the words related to the shift and OH group were deleted in the manuscript.
In Fig.3, they wrote that ZnO nanorods were wrapped. However, it is not clear. I think the ZnO nanorods became small and changed to particle shape. Please estimate supecific surface area of each samples. It is important to compare the photocatalitic activites.
Response: Thank you very much for your comment and suggestion. As shown in HR-TEM image of Figure 3d, the crystalline ZnO were covered by low-crystalline g-C3N4. According to SEM images in Figure 2 and XRD patterns, it is believed that the nanorod-like structure were retained for ZnO in these nanocomposites. The specific surface area were studied and added in Supporting Information.
About PL, which is the exitation wavelength? I think the shapes of the PL spectra of ZnO and composites are strange. Please check the excitation wavelenght.
Response: Thank you for your question. The excitation wavelength is 365 nm. We tested ZnO and the nanocomposites again, and the spectra were almost the same as that in Figure 5d of this manuscript.
In XPS spectra, the baselines of O1s and Zn 2p look like incollect. In O1s, there are three peaks as same as those of the coposites.
Response: Thank you very much for your comment. The XPS data were re-analyzed and highlighted in Figure 6. In the rectified figure, O 1s were fitted with three peaks in this manuscript.
The desctription of the Fig.7 is not enough. Please add the information.
Response: Thank you for your advice. We have added the information and highlighted in manuscript.
The reason why 0.1:1 CN samples are best photoreactivity is not clear. Please discuss more based on the results.
Response: Thank you very much for your suggestion. According to the results, we added some explanation and highlighted in the revised manuscript.
Reviewer 2 Report
Authors prepared g-C3N4/ZnO nanocomposites and tested photocatalytic activity for MB.
However, there is no scientific depth and new important finding in the present results.
This reviewer cannot recommend publication.
Author Response
Comments and Suggestions for Authors
Authors prepared g-C3N4/ZnO nanocomposites and tested photocatalytic activity for MB.
However, there is no scientific depth and new important finding in the present results.
This reviewer cannot recommend publication.
Response: Thank you very much for your comment. It is commonly believed that the activity of nanocomposite will be enhanced as long as the band positions match well with each other, in comparison of the correspond single-component photocatalyst. However, we have encountered the opposite experimental results more than one time. To the best of our knowledge, there are few papers reported this experimental phenomenon. In this manuscript, we are trying to study the reasons for this phenomenon through g-C3N4/ZnO nanocomposites. There are still a lot to do to excavate the mechanisms, and this work might pave avenue for effective nanocomposite design and synthesis.
Reviewer 3 Report
Authors described the in-situ synthesis of C3N4/ZnO nanocomposites. They preprared the nanocomposites by two different ways and compared them in respect to the physical properties and photocatalytic activities. The work is original and the manuscript is well organized. However, there are a few issues that need to be revised. Therefore, this manuscript is acceptable after minor revision.
In the abstract session, authors used the terms: x:1 and x:1 CN without explanations. Please add meanings of the terms in the abstract session.
In the abstract session, authors mentioned MB photodecomposition unver UV and visible light. However, the experiments were done under UV and sunlight. Please revise this point.
In the abstract session, reaction rate of 2.01 min-1 was mentioned but it is "reaction rate constant".
Please include the source of sunlight in the methods
In the session, 1.1, it is not clear what is "an appropriate dose of urea". Please indicate the used urea amunts in grams for all samples.
in "x:1", does x means the added amount or product amount? please define x.
Typically the C3N4 produced by thermal decompositon gives very low yield and the yield is quite low and affected by the presence of foreign elements during the decompositon. Therefore it is likely that even the same ratio of zinc acetate and urea was used for the two reaction, e,g. x:1 vs. x:1 CN, different amounts of C3N4 might be obtained. authors described many physical characterizations but none of them provides quantitative information about x and ZnO. Please include this information by (for example) thermographic ananlysis (TGA) since it is important to estimate the photocatalytic activity of each composites.
Authors nicely presented the scavenger experiments. Please add a scheme of photocatalysis with a possible bandgap alignment.
In Fig 3 caption, (e) is missing. Also the elemental mapping is by Energy-dispersive X-ray spectroscopy (EDS) images. In page 4, line 161, reivse "EDX spectra" and to "EDS images" and give a full name of it. (also in the methods, add its instrumental infomation)
In page5, line 163, authors mentioned about sonication for 2h. Please add the evidence of no change of morphology in the supplementary info.
In page 7, line 217, the peak corresponding to oxygen defects was mentioned with a reference. However I am puzzled how oxygen peak can be shown by XPS if it is a vacancy site? please explain it.
The conclusion is rather too short. Please include explanation about energy transfer between the two components.
In page 3, line 96, "thermally decomposing" should be "thermally decomposed"
In page 4, line 139. the wording of "that might because" needs to be revised.
Please check English again for any possible errors.
Author Response
Authors described the in-situ synthesis of C3N4/ZnO nanocomposites. They preprared the nanocomposites by two different ways and compared them in respect to the physical properties and photocatalytic activities. The work is original and the manuscript is well organized. However there are a few issues that need to be revised. Therefore, this manuscript is acceptable after minor revision.
Response: Thank you very much for your kind comment.
In the abstract session, authors used the terms: x:1 and x:1 CN without explanations. Please add meanings of the terms in the abstract session.
Response: Thank you very much for your kind comment and suggestion. We have added the meanings of these two terms and highlighted in the abstract session in manuscript.
In the abstract session, authors mentioned MB photodecomposition under UV and visible light. However, the experiments were done under UV and sunlight. Please revise this point.
Response: Thank you a lot for your advice. We have revised the point and highlighted in manuscript.
In the abstract session, reaction rate of 2.01 min-1 was mentioned but it is "reaction rate constant".
Response: Thank you for your comment. We have added the word “constant” in the abstract part.
Please include the source of sunlight in the methods.
Response: Thank you very much for your suggestion. The experiments were executed just outside laboratory. Therefore, we added and highlighted the place location information in the method part.
In the session, 1.1, it is not clear what is "an appropriate dose of urea". Please indicate the used urea amounts in grams for all samples.
Response: Thank you very much for your interesting question. In our lab, 20 grams of urea produced 0.8 gram graphitic carbon nitride during thermal decomposition. Therefore, it is easy to calculate the amounts of urea added for each sample.
In "x:1", does x means the added amount or product amount? Please define x.
Response: Thank you for your question. The x has been defined in line 74 as you can see above Scheme 1.
Authors nicely presented the scavenger experiments. Please add a scheme of photocatalysis with a possible bandgap alignment.
Response: Thank you very much for your kind suggestion. The charge carriers’ separation and transfer were truly important in photocatalytic reaction. However, in this manuscript, we focused on the mayor active species for methyl blue photodegradation.
In Fig 3 caption, (e) is missing. Also the elemental mapping is by Energy-dispersive X-ray spectroscopy (EDS) images. In page 4, line 161, revise "EDX spectra" and to "EDS images" and give a full name of it. (also in the methods, add its instrumental infomation)
Response: Thank you very much your helpful suggestion. We have revised these mistakes and highlighted in the manuscript.
In page 5, line 163, authors mentioned about sonication for 2h. Please add the evidence of no change of morphology in the supplementary info.
Response: Thank you for your kind suggestion. We have studied the morphology of ultrasonically treated 0.1:1 CN nanocomposite by X-ray powder diffraction, and the patterns were shown in Figure S3.
In page 7, line 217, the peak corresponding to oxygen defects was mentioned with a reference. However I am puzzled how oxygen peak can be shown by XPS if it is a vacancy site? Please explain it.
Response: Thank you for your question. This refers to the oxygen in deficient regions of ZnO nanorods. In order to avoid confusion, we have changed and highlighted the description as “oxygen in deficient regions”.
The conclusion is rather too short. Please include explanation about energy transfer between the two components.
Response: Thank you very much for your suggestion. Generally, the conclusion should be small and concise. We insisted not to add the “energy transfer between the two components” since we did not mention this content in our work. We really feel terribly sorry for that.
In page 3, line 96, "thermally decomposing" should be "thermally decomposed"
In page 4, line 139. The wording of "that might because" needs to be revised.
Please check English again for any possible errors.
Response: Thank you very much for your kind reminding. We have modified these grammar faults and highlighted in the manuscript. We also checked the manuscript to make sure that there are no other errors in this paper.
Round 2
Reviewer 2 Report
Zhang et al prepared a series of g-C3N4/ZnO nanocomposites and evaluated in Methylene blue dye photodecomposition under UV and sunlight irradiation. They claimed that g-C3N4/ZnO (0.1:1 CN) showed the best photocatalytic activity among all g-C3N4/ZnO (x:1 CN) 249 nanocomposites. However, no significant improvement in activity is seen in Fig 7. There is no significant role of g-C3N4. In Figure 1, no XRD peak of g-C3N4 was found. Authors should provide sufficient experimental data and detailed Figure caption.
Author Response
Zhang et al prepared a series of g-C3N4/ZnO nanocomposites and evaluated in Methylene blue dye photodecomposition under UV and sunlight irradiation. They claimed that g-C3N4/ZnO (0.1:1 CN) showed the best photocatalytic activity among all g-C3N4/ZnO (x:1 CN) nanocomposites. However, no significant improvement in activity is seen in Fig 7. There is no significant role of g-C3N4. In Figure 1, no XRD peak of g-C3N4 was found. Authors should provide sufficient experimental data and detailed Figure caption.
Response: Thank you very much for your comments and suggestion. In this manuscript, two series of g-C3N4/ZnO were synthesized through two different synthesis procedures. The photocatalytic activities in methyl blue photodegradation of these prepared samples were also studied under ultraviolet light and sunlight irradiation, respectively. As can be seen in Fig. 7, all the samples showed enhanced activity under sunlight irradiation and relatively low activity under ultraviolet light irradiation, which might mainly due to low power of the lamp used in this manuscript.
Forty minutes are required to decompose methyl blue completely under sunlight irradiation, while only fifteen minutes are needed to photo-decompose the solution in the same system. Furthermore, for this pseudo first order photodecomposition, the reaction constant could reflect the samples’ activity directly. Kinetics study showed that a maximum reaction rate constant of 0.1862 min-1 was obtained over 0.1:1 CN photocatalyst under solar light irradiation, almost two time higher than that of bare ZnO, which can be ascribed to enhanced visible light sensitivity (shown in Figure 5a and 5b), suppressed recombination of charge carriers (can be seen in Figure 5d) of the nanocomposite with g-C3N4 introduction.
In Figure 1, the XRD patterns for g-C3N4 can be detected when the content is high enough in these x:1 nanocomposites (shown in Figure 1a). However, among these x:1 CN photocatalysts, g-C3N4 cannot be found in XRD patterns, which mainly due to the high dispersion of g-C3N4 in the nanocomposites, as demonstrated by element mappings in Figure 3e.
English expression is rechecked and polished in this revised manuscript.